# Influence of *Funneliformis mosseae* enhanced with titanium dioxide nanoparticles (TiO$_2$NPs) on *Phaseolus vulgaris* L. under salinity stress

**Nashwa El-Gazzar**[1]*, **Khalid Almaary**[2], **Ahmed Ismail**[3], **Giancarlo Polizzi**[4]

**1** Botany and Microbiology Department, Faculty of Science, Zagazig University, Zagazig, Egypt,
**2** Department of Botany and Microbiology, College of Science, King Saud University, Riyadh, Saudi Arabia,
**3** Plant Pathology Research Institute, Agricultural Research Center, Giza, Egypt, **4** Dipartimento di Gestionedei Sistemi Agroalimentari e Ambientali, Sezione di Patologia Vegetale, Catania, Italy

* mora_sola1212@yahoo.com

**Data Availability Statement:** All relevant data are within the manuscript and Supporting Information files.

## Abstract

The Arbuscular mycorrhizal fungi (AMF) (**Funneliformis mosseae**), are the most widely distributed symbiont assisting plants to overcome counteractive environmental conditions. In order to improve the sustainability and the activity of AMF, the use of nanotechnology was important. The main objective of this study was to investigate the effect of titanium dioxide nanoparticles (TiO$_2$NPs) on the activity of AMF in common bean roots as well as its activity under salinity stress using morphological and molecular methods. The activity of AMF colonization has increased in the presence of TiO$_2$NPs especially for arbuscule activity (A%), which increased three times with the presence of TiO$_2$NPs. The improvement rate of **Funneliformis mosseae** on plant growth increased from 180% to 224% of control at the lowest level of salinity and increased from 48% to 130% at higher salinity level, respectively. The AMF dependencies for plant dry biomass increased in the presence of TiO$_2$NPs from 277% in the absence of salinity to 465 and 883% % at low and high salinity levels, respectively. The presence of AMF co-inoculated with TiO$_2$NPs resulted in increasing the salinity tolerance of plants at all levels and reached 110% at salinity level of 100 mM NaCl. Quantitative colonization methods showed that the molecular intensity ratio and the relative density of paired inocula AMF Nest (NS) or chitin synthases gene (Chs) with TiO$_2$NPs were higher significantly P.>0.05 than single inoculants of AMF gene in roots under the presence or the absence of salinity by about two folds and about 40%. Hence, the positive effect of TiO$_2$NPs was confined to its effect on AMF not on bean plants itself.

## Introduction

Soil salinity is one of the detrimental 'biotic stresses'; which influences plant growth and crop production worldwide. At the physiological scale, soil salinity reverses ionic and water homeostasis [1]. However, at a cellular level, it leads to boosted accumulation of reactive oxygen species (ROS). The raised ROS level adversely affects the cellular redox homeostasis, consequently

**Funding:** This work was funded by King Saud University, Deanship of Scientific Research College of Science Research Center and Zagazig University, Egypt, Faculty of Science. We are also indebted to Zagazig University, Egypt for facilities. The funders had no role in study design, data collection and analysis, decision to publish, or preparation of the manuscript.

**Competing interests:** The authors have declared that no competing interests exist.

commanding with "oxidative damage". Sundry molecular regulators can be applied to broaden salt-tolerant genotypes [2]. Therefore, development of crop production in saline soils is very necessary; soil salinity is one of the major problems for agriculture in Egypt as Egyptian lands are subjected to extreme climatic factors such as high temperatures and drought. Under these conditions, dissolved salts may accumulate in soils because of the insufficient leaching of ions. Only few crop species are adapted to saline conditions. Hence, studies aiming at development of plant agriculture in saline soils are needed [3]. It is obvious that there is a need to continue research to find out non-traditional and novel protocols for overcoming salt stress; plants have evolved complex mechanisms that contribute to the adaptation to salinity conditions [4]. This study is an endeavor to improve the propagation of the plant *Phaseolus vulgaris* under salinity conditions by both, *Funneliformis mosseae* and $TiO_2NPs$.

Arbuscular Mycorrhizal fungi are used to improve plant propagation in saline soils. Arbuscular mycorrhizal symbiosis is known as the mother of the plant roots in natural ecosystems and enables the plants to cope with a biotic and biotic stresses [5]. AMF have been reported to be involved in nutrient use efficiency, photosynthesis and plant metabolism [6]. The symbiosis with AMF can ameliorate the plants response to salinity and have beneficial effects on plant growth and yield; this symbiosis was reported to reduce Na+ uptake and translocation and induce the uptake of K+, Ca+ and Mg+2 and stimulate the K+/Na+ and Mg+2/Na+ ratios in shoots [7].

Nanotechnology is used recently to overcome many agricultural problems. Metalic nanoparticles enhance the seed germination, shot/root growth, biomass production and physiological activates [8]. In a previous study [9], it was showed that uptake of cucumber plants by 0 to 750 mg/Kg $TiO_2NPs$ increased the chlorophyll content of leaves; the cucumber fruits obtained after 150 days of cultivation were found to be richer with 35% potassium and 34% more phosphorus compared with the fruits from the control plants. It was found that $TiO_2NPs$ (500mg/L) improved the nitrates absorption by many plants and favor the conversion of inorganic nitrogen to organic nitrogen in the form of protein and chlorophyll which lead to an increase in the fresh weight and dry weight of the plant [10].

The recent perspectives have been raised that $TiO_2NPs$ may positively affect crops and soil microbial communities, including beneficial microbes such as AMF [11]. In addition, it was proved that nanoparticles have positive effects on microbial communities and the ecosystem services [12,13]. Also, previous studies have simultaneously examined the effects of NPs on both plant growth and rhizosphere microbial communities [14]. Previous literature confirmed that the application of nanoparticles at concentrations of 5, 25 and 50 ppm was found to stimulate the formation of mycorrhizae in seedlings of pedunculate oak, with the highest effect at intermediate concentrations (25 ppm) [15]. However, the influence of nanoparticles on plant growth and rhizospheric microbial communities, especially the important symbiotic microbes, such as arbuscular mycorrhizal (AM)fungi, remains under debate [16].Thus, considering its multifaceted effects, precise optimizations of doses and modes of $TiO_2NPs$ applications are required, before its applications could be recommended at a wider scale on crop plants [17]. In addition, further studies with primers that specifically target AMF are required [18].

In a previous study that carried out previously [19] using the same experimental mycorrhizal strain tested (AMF) investigated the effect of such mycorrhiza fungus to red kidney and wheat plants in enhancement their activity to heavy metal-polluted soil. The mycorrhizal plants showed increasing in their dry weight, height, chlorophyll, sugar and protein content, nitrogen and phosphorus efficiencies, leaf conductivity, transpiration rate, nitrogenous, acid and alkaline phosphates activities of all plants irrespective of the presence of heavy metal. To extend and develop the role of this experimental mycorrhiza used the enhancement plant production; the effect of this AMF on *Phaseolus vulgaris* grown in saline soils was studied either

singly or in combinations with $TiO_2NPS$ using morphological, physiological and molecular methods through the detection of intensity and relative density of nest and ChS genes in the *Funneliforms mosseae*.

## Materials and methods

### Source of mycorrhiza, pot and potting mixture

The AMF used in this study was used in a previous study that carried out in our laboratory of Mycology, Botany Department, Faculty of Science, Zagazig University [19]. This AMF was originally provided by Prof. Dr. Gamal Abdel Fattah Ouf, Faculty of Science, Mansoura University, Egypt. It was stored at 20 $^{o}$C as colonized root fragment in sandy loamy soil sample taken from the rhizosphere region of the plant guania grass (*Megathyrsus maximus*) in plastic bag (250/500 cm$^3$ bag) (Gomhuria Comp., Cairo, Egypt). To stimulate the AMF, five pots, each containing 950g sandy loamy soil were prepared and each soil fraction was mixed 50g soil that contained AMF and cultivated with *Megathyrsus maximus* irrigated with water every one week for two months [20]. The shoot system was as cut and the remained root systems were mixed with the soil and about 10g aliquots were used as an inoculum for further experiments.

### Soil and mature seeds of the examined plant

Sandy loamy soil samples (1 kg aliquots in 2000 cm$^3$ bags disposable) (Gomhuria Company. Cairo, Egypt) were collected from reclaimed soil at (0–20 cm) in the Sharkia province, Egypt and autoclaved twice at 121˚C (15-lbs/in$^2$) for one hour. These serial soil samples were put in serial pots that were sterilized by UV-radiation chamber for 12h. The soil was analysed as follows: physical analysis was done as described by [21], sand 80%, silt 15% and clay 5%. Chemical analysis was done according to [21,22]as nitrogen 0.85g/Kg, potassium 0.186g / kg, phosphorus 0.32g / kg and a pH at (7.9).

Seeds of a local variety of common bean (*Phaseolus vulgaris* L.) were obtained from Agronomy Department, Agriculture Research Centre, Giza, Egypt. They were surface sterilized with 0.01% $HgCl_2$, washed 3–4 times with distilled water and sown in black plastic pots containing 2000 g of soil [23].

### Biosynthesis and characterization of $TiO_2NPs$

Biosynthesis of $TiO_2NPs$ was produced using *Aspergillus flavus* KF946095 [24], following the method described in the literature [25]. Transmission Electron Microscope (TEM)was used for characterization of $TiO_2NPs$ on Joel 1230 operated at 100KV connected with CD camera, Japan. TEM analysis was done at National Research Centre, Giza, Egypt. Sizes of $TiO_2NPs$were determined using Zeta sizer Nano series (Nano ZS) (Malvern, UK), at Regional Center for Food and Feed, ARC, Giza, Egypt. Fourier Transform Infrared Spectroscopy (FTIR) (Thermo Nicolet model 6700 spectrum) detected the presence of functional groups of amino acids as protecting agents to $TiO_2NPs$ Located at Micro-analytical Center, Cairo University, Giza, Egypt [26].

### Preparation of biosynthesized $TiO_2NPs$ solution and salt solutions

The biosynthesized $TiO_2NPs$100mg /kg of soil was dissolved in Tris-HCl buffer (pH = 8),and the solution was stirred vigorously and suspended in 400 mL of Milli-Q deionized water by sonication in a water bath for 30 min at 30˚C with occasional stirring [27,28]. The solutions were then centrifuged at 8500 rpm for 30 min and then, washed with water before dispersing in water (pH = 7). $TiO_2NPs$ suspensions were applied twice after seed germination [29]. Salt

solutions were applied after seed germination at concentrations of about 0, 100 and 200 mM of NaCl. Control treatments were irrigated with dist de-ionized water.

## Experimental design and layout

The experiment was carried on 6 replications from each treatment. The treatments are given in Table 1. Treatments and controls were allowed for growth of *Phaseolus vulgarius* for 3 months.

Seeds of common beans were sown in plastic pots containing soil and were thinned to 5 plants per pot after one week of germination. The plastic pots were placed in a growth chamber at 25/20˚C day/night, 11 h day, with 60–70% of relative humidity and water to 75% of WHC 2 times week. The plants were harvested 90 days after sowing date. Observations were recorded at 45 and 90 days of inoculation as the following:

## AMF (colonization and mycorrhizal dependency)

The whole plant was used for biomass measurement after 45 and 90 days of sowing. The plant samples were washed with distilled water to remove the adhering soil particles and separated from the above-ground parts and the root parts. Firstly, samples were air-dried in cool and well ventilating places. The weighed subsamples of fresh roots and leaves were used for root mycorrhizal colonization by morphological and molecular methods and acid and alkaline phosphatase activity assessment [30]. Root samples were collected at an interval of 45 days, AMF colonization of root was examined after staining with 0.5% trypan blue in lactophenol [31] method. Root segments, each approximately 1-cm long, were selected at random from a stained sample and mounted on microscopic slides. Slides with stained root segments were carefully observed using the microscope under suitable magnification. Percent Mycorrhizal root colonization (Frequency of mycorrhizal infection F%, Mycorrhizal intensity M% and Arbuscular activity A%) was determined by gridline intersect slide method [32]. Spores produced by AMF in the root zone soil were estimated by wet sieving and decanting method [20]. The mycorrhizal dependency (MD) and of plants were calculated as described previously [33].

## Acid &Alkaline phosphatase activity analysis

Fresh leaves were harvested from the selected plants treated with various amendments for leaf estimation of enzymes including, acid and alkaline phosphatase [34] at 45th and 90th day of experimentation.

**Table 1. Different treatments allowed for growth of *Phaseolus vulgarius*.**

|   | Treatments |   |
|---|---|---|
| 1 | common bean control (P) | non-inoculated with *F mosseae* (NM) |
|   |   | Inoculated with *F. mosseae* (M) |
| 2 | TiO$_2$NPs | non-inoculated with *F mosseae* (NM) |
|   |   | Inoculated with *F. mosseae* (M) |
| 3 | NaCl 100mM | non-inoculated with *F mosseae* (NM) |
|   |   | Inoculated with *F. mosseae* (M) |
| 4 | NaCl 200mM | non-inoculated with *F mosseae* (NM) |
|   |   | Inoculated with *F. mosseae* (M) |
| 5 | TiO$_2$NPs + NaCl 100mM | non-inoculated with *F mosseae* (NM) |
|   |   | Inoculated with *F. mosseae* (M) |
| 6 | TiO$_2$NPs + NaCl 200mM | non-inoculated with *F mosseae* (NM) |
|   |   | Inoculated with *F. mosseae* (M) |

## Dry biomass and protein analysis

Plant samples were air-dried in cool and well ventilating places, then they were oven dried (68°C ±2°C, 24 h). The dry weight of each part was weighed to calculate the biomass. In addition, a known weight of fresh leaf or root tissues was used for soluble protein analysis according to [26,35,36].

## Plant macronutrient contents analysis

The total (P) of shoot and root dried plant samples was determined by the method described previously [37]. Minerals content in dried samples were determined by the method of [38], potassium ($K^+$), and sodium ($Na^+$) were determined spectrophotometrically in the acid digested samples by using atomic absorption spectrophotometer (Model Unicam 969). Total N was measured in the shoot and root samples of 0.1 g dry mass using the Kjeldahl method [39]. DNA extraction, detection of AMF in plant roots by nested and Chitin synthase, Polymerase chain reaction (PCR)

## DNA isolation

DNA was extracted from 0.5 to 1.0 g of fresh roots according to the modified method of Dellaports et al. [40]. Plant roots were grinded to powder using liquid nitrogen, re-suspended in extraction buffer (6.25 mM potassium ethyl xanthate, 100 mM Tris-HCl, pH 7.5, 700 mM NaCl, 10 mM EDTA, pH 8) and incubated at 65°C for 1 h. Plant and fungal DNA was extracted (1v:1v) with phenol: chloroform: isoamyl alcohol (25:24:1) and once with chloroform: isoamyl alcohol (24:1) [41]. DNA was precipitated with isopropanol (1v:1v) and 3M sodium acetate pH 5.2 (1v:1/10v) for 1 h at room temperature and centrifuged at 12000 rpm for 25 min. The pellet was washed with 70% ethanol, dried under vacuum (Concentrator 5301 Eppendorf, Hamburg, Germany) and re-suspended in 50 μl of ultrapure water and stored at 20°C for further use.

## PCR amplification for screening the potential activity, intensity and relative density of mycorrhizal genes

The genes coding for NS1 and NS8 and genes coding for ChS F and R was used as molecular markers to screen mycorrhizal gene activity and intensity as described previously [42]. Based on the conserved sequence of the NS1 gene, Oligo primers NS1-F (5-GTAGTCATATGCTT GTCTC-3) and NS8-R (5-TCCGCA GGTTCACCTACGGA-3) that were designed and synthesized as described previously [43]. The ChS gene primers, OligoChS-F were (5-CTC AAG CTT ACT ATG TAT AAT GAG GAT-3) and ChS-R were (5-GTT CTC GAG TTT GTA TTC GAA GTT CTG-3) according to [44]. The fungal isolates were initially screened by PCR for the presence of the nested gene. PCR amplification was carried out using the primers NS1-F and NS8-R in a 25ll reaction mixture. The PCR reaction mixture contains 10 μl of 2× PCR master mixture (i-Taq™, Cat. # 25027, INTRON Biotech), 2 μl of (10 ng) DNA, 1 μl of forward and reverse primers (10 pmol/μl) and the mix was completed to 20 μl with sterile distilled water. PCR amplification was performed at Thermal Cycler 006 (A&E Lab Co. Ltd. England), programmed to initial denaturing step at 94°C for 2 min, denaturing at 94°C for 20 sec, annealing at 55°C for 30 sec, extension at72°C for 1 min, for 35 cycles and final extension for 5 min at 72°C. The PCR products were analyzed by 1.5% agarose gel, visualized by gel documentation [41].

## Gel electrophoresis

Genomic DNA samples, as well as nesting PCR amplification products, were separated on 1.0% (w/v) agarose gel. Agarose gel 1.5% (w/v) was used for nesting PCR amplification products comparing to the DNA ladder (1 kb Nex-gene Ladder, Puregene). The negative control PCR reactions without fungal g DNA were used. The amplicons were visualized by gel documentation system (VilberLourmat, France). Electrophoresis was carried out and visualized under 300 nm UV light.

## Amplified genes activity

Electrophoresis amplicons were documented using an imaging system (Molecular Imager, Gel-Doc XR, BIORAD, USA) for detecting the genes expression activity (Intensity & Relative density). DNA profiles were acquired with the Image Master VDS system and analyzed using the Image J Master 1D Elite software (Amersham Bioscience, ColognoMonzese, Italy).Quantities and intensity of genes after PCR productwere measured according to the following equation; Relative expression of NS genes = Area (Mean) of the gene of treated/ control = x fold ratio. By this way, arbuscular mycorrhiza relative density (AMRD) was calculated for each treatment in roots as follows: AMRDR (root) = (Molecular ratio of each fungus in root x root dry weight) [45].

## Statistical analysis

The differences between means were compared using Fisher's least significant differences test (LSD) with the statistical WASP software version 2.0; LSD: At significant level (P>0.05).Sample symbols (a.a): mean non significant difference (a.b): mean significant difference [46]. Principal component analysis (PCA) was performed on measured parameters of common bean plants under salinity stress in response to treatments with AMF and $TiO_2$NPsData of growth, physiological parameters and mycorrhiza levels in common bean plants inoculated with AM *Funneliformis mosseae* and $TiO_2$NPs in the presence of salinity were subjected toanalysis of variance (ANOVA) [47].

## Results

The results exhibited that the morphology of biologically synthesized $TiO_2$NPs by *A. flavus*KF946095 were oval, cubic and rod-shaped (Fig 1A) with diameter of about 68 nm (Fig 1B).To further validate the presence of stabilizing proteins, $TiO_2$NPs were analyzed by FTIR, that confirmed the presence of various functional groups at 1633.41cm$^{-1}$, 1458.89cm$^{-1}$, 1078.51cm$^{-1}$, 3430.74cm$^{-1}$and 2366.23cm$^{-1}$corresponding to carbonyl residues and peptide bonds of proteins responsible for the synthesis of the $TiO_2$NPs(Fig 1C). The peaks were classified along with the functional groups (Table 2).

Data presented in Table 3 showed that frequency of mycorrhizal root segments (F %), the intensity of mycorrhizal colonization in root tissues (M %) and arbuscule frequency in root systems (A %) increased in the response to $TiO_2$NPs irrespective to presence and absence of salinity stress. The values of F%, M% and A% increased from 87, 31 and10.9% in absence of $TiO_2$NPs into 96, 45.5 and21.8% (P> 0.05) respectively in the presence of $TiO_2$NPs.

The data also showed that the mycorrhizal association with common bean plants was significantly influenced by salinity. The mycorrhizal infection was reduced from 87% to 59% and 47% at low and high salinity respectively in the absence of $TiO_2$NPs and from 96% to 90% and 84% at the same levels in the presence of $TiO_2$NPs.The presence of salinity reduced mycorrhizal intensity (M %) by about 60% and 66% (P> 0.05) at low and high salinity respectively in

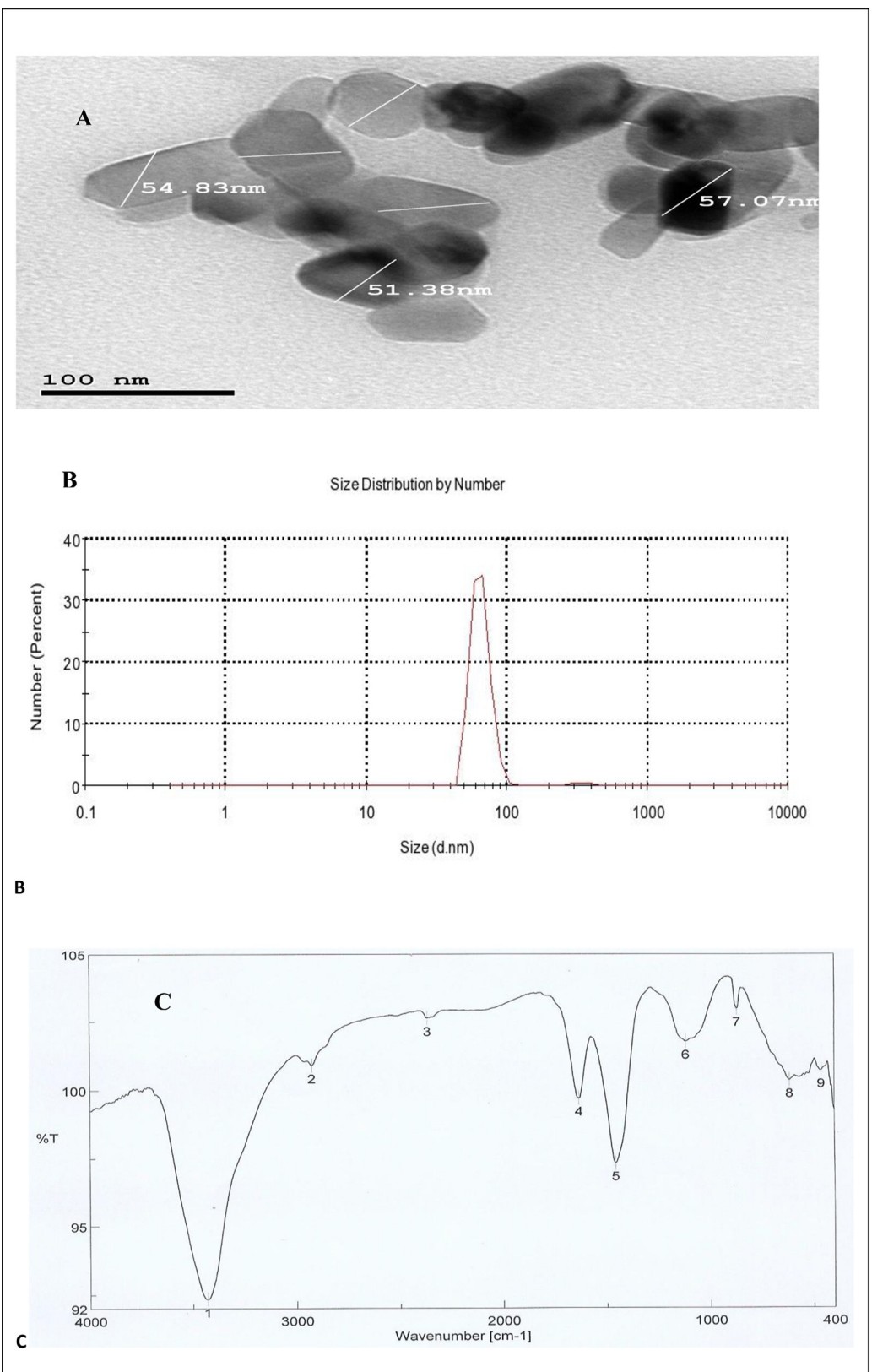

**Fig 1.** Characterization of $TiO_2NPs$: (A) TEM of $TiO_2NPs$ with oval, cubic and rod-shaped, (B) Zeta sizer of $TiO_2NPs$ at 68nm and (C) FTIR of $TiO_2NPS$.

the absence of $TiO_2NPs$.This reduction was significantly inhibited in the presence of $TiO_2NPs$ reaching to 35% and53% (P> 0.05) for two levels of salinies (low and high) respectively. Arbuscule frequency in root systems (A %) exhibited similar trends where the A % was greatly reduced by about 45% and 64% of its activity under the two levels of salinity in absence of $TiO_2NPs$.While this reduction was significantly inhibited to 23% and51% at the two salinity levels in the presence of $TiO_2NPs$. On the other hand, the results of mycorrhiza development at the end of the experiment (90 days) were higher than that at 45 days in the presence of $TiO_2NPs$ especially in the presence of salinity.

The data in Table 3 indicated that mycorrhizal dependencies for plant dry mass increased by raising salinity. The increase varied from 254% in absence of salinity into 369 and358%(P> 0.05) at the two salinity levels, respectively in the absence of $TiO_2NPs$. While in the presence of $TiO_2NPs$ the increase varied from 277% in the absence of salinity to 465 and883% % (P> 0.05) at low and high salinity levels, respectively.

The results obtained in this study, (Fig 2A–2E) showed a mature vesicular at 90 days, AMF around and inside common bean plants are roots, intraradical mycelium with vesicular and symbiotic phase of *Funneliformis mosseae* in common bean plants roots with $TiO_2NPs$, respectively. Colonization of AMF in roots of common bean plants in the presence and the absence of salinity stress showed in (Fig 3A–3E).

The results given in Table 4 showed that the dry weight of common bean plants colonized by *Funneliformis mosseae* was higher than non-mycorrhizal plants either in normal or saline conditions. The results also showed that the presence of *Funneliformis mosseae* improved the growth of the plant, both in the normal or saline conditions, where the improvement rate was 154% (P>0.05) than the control (Non AMF plants) in normal circumstances. While, at a low level of salinity, the presence of AMF was able to remove the negative effect of salinity. The results showed that the negative effect of salinity at a low level (-24% of control) in the absence of AMF was reversed in the presence of AMF. Where, the salinity effect alleviated and the growth rate improved by 180% (P>0.05) in relative to control. Also, at higher level of salinity, the growth increased from– 59% into 48% when compared to control. The results given in Table 4 also indicated that AMF supplemented with $TiO_2NPs$ increased the rate of plant growth, regardless of the presence of salinity or not, where the improvement rate increased from 154% to 195% than that in control in the absence of salinity. While the improvement rate increased from 180% to 224% (P>0.05) at the lowest level of salinity and increased from 48%

**Table 2. Classification of FTIR beaks of $TiO_2NPS$.**

|  | FTIR peaks ($cm^{-1}$) | Classification |
|---|---|---|
| 1 | 3430.74 | N-H stretching (aliphatic amine) peptide bonds of proteins and carbonyl residues |
| 2 | 2925.48 | C-H stretching alkyne |
| 3 | 2366.23 | O = C = O carbon dioxide |
| 4 | 1633.41 | C = C stretching alkene, carbonyl residues |
| 5 | 1458.89 | C-H alkane, carbonyl residues and peptide bonds of proteins |
| 6 | 1118.51 | C-O stretching secondary alcohol mononuclear aromatics and carbonyl residues |
| 7 | 870.703 | C-H Bending |
| 8 | 615.181 | C = C alkene |
| 9 | 462.832 | C = C alkene |

**Table 3. Effect of salinity levels on AMF development (as indicated by trypan blue staining) and Mycorrhizal Dependency (MD) of mycorrhizal, and non-mycorrhizal plants supplemented with and without TiO₂NPs.**

| Periods (days) | | F% | | M% | | A% | | MD |
|---|---|---|---|---|---|---|---|---|
| Treatments | | 45 | 90 | 45 | 90 | 45 | 90 | 90 |
| P | Nm | - | - | - | - | - | - | |
| | M | 85 | 87 | 34.3 | 31.85 | 10.8 | 10.9 | 254 |
| TiO₂NPs | Nm | - | - | - | - | - | - | |
| | M | 65 | 96 | 32 | 45.4 | 12.6 | 21.8 | 277 |
| NaCl 100mM | Nm | - | - | - | - | - | - | - |
| | M | 78 | 59 | 15.2 | 12.8 | 6.1 | 5.95 | 369 |
| NaCl 200mM | Nm | - | - | - | - | - | - | - |
| | M | 70 | 47 | 13.2 | 10.8 | 4.5 | 3.95 | 358 |
| TiO₂NPs + NaCl 100mM | Nm | - | - | - | - | - | - | - |
| | M | 83 | 90 | 20.3 | 29.6 | 9.6 | 16.8 | 465 |
| TiO₂NPs + NaCl 200mM | Nm | - | - | - | - | - | - | - |
| | M | 80 | 84 | 15.7 | 21.3 | 6.8 | 10.7 | 883 |

F% Frequency of mycorrhizal root segments, M% intensity of mycorrhizal colonization, A% arbuscule frequency in root systems. NM; Non-mycorrhizal plants and M; Mycorrhizal plants.

to 130% at the highest salinity level. By examining the results illustrated in Table 4, it was found that the rate of improvement of plant growth in the saline environment in the presence of mycorrhiza and TiO₂NPs is better than that obtained by mycorrhiza alone. While, TiO₂NPs alone do not have a significant effect on the plant growth in this conditions. These results

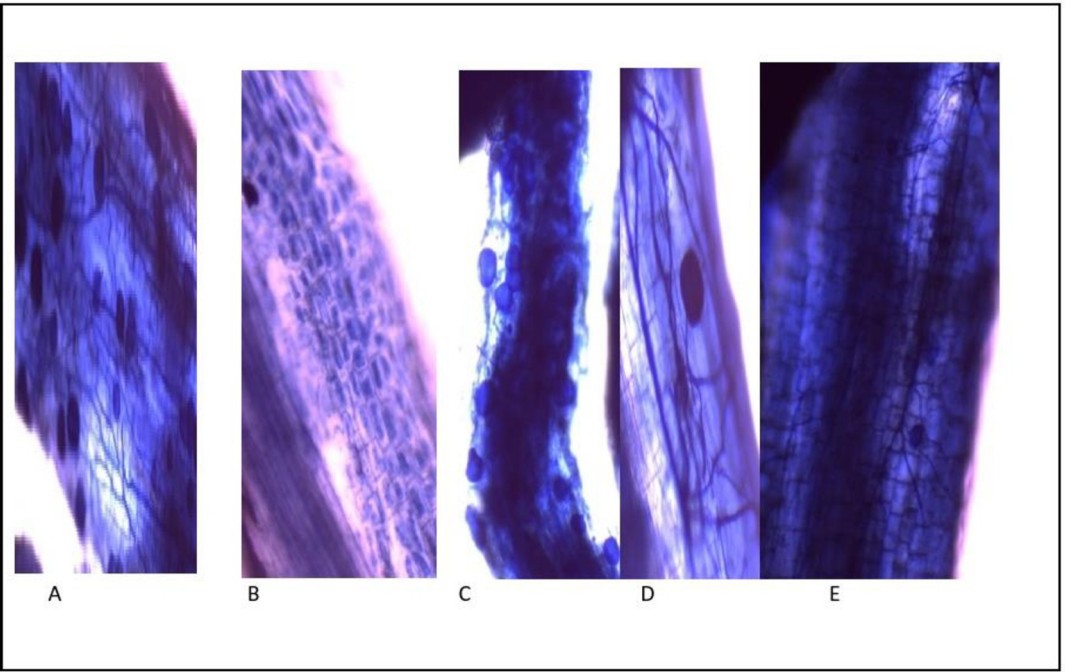

**Fig 2.** Inoculation of common bean plants roots with AMF and TiO₂NPs showing: (A) mature vesicles at 90 days; (B) Arbuscular; (C) vesicles of AMF around and inside roots; (D) intraradical mycelium with vesicles; (E) symbiotic phase of AMF in roots.

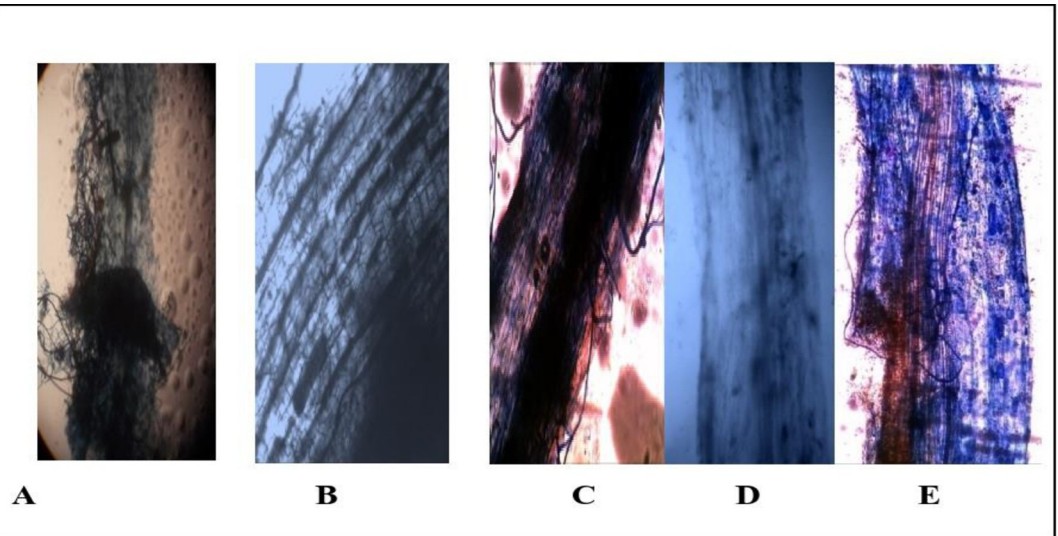

**Fig 3.** Colonization of AMF in roots of common bean plants in the presence and the absence of salinity stress: (A) mature arbuscule at 90 days; (B) arbuscular; (C) symbiotic phase of AM in roots (20X); (D) intraradical mycelium; (E) arbuscule of AMF under salinity stress at 90 days; (F) arbuscule, vesicles and intraradical mycelium of AMF with TiO$_2$NPs under salinity stress.

indicated that the growth improvement in the common bean plants in the saline environment were due to the effect of TiO$_2$NPs on the activity of mycorrhiza and not on the plant itself, suggesting that the effectiveness of TiO$_2$NPs in mycorrhizal symbiosis.

The data recorded in Table 4 revealed that the tolerance indexes of common bean plants were significantly decreased(P>0.05) by increasing the salinity levels. The maximum reduction was obtained in non-mycorrhizal plants treated with and without TiO$_2$NPs, while it was

**Table 4. Effect of salinity levels on dry wt, TI, efficiency of inoculants and TiO$_2$NPs concentration of mycorrhizal, and non-mycorrhizal plants supplemented with and without TiO$_2$NPs.**

| Periods | | Dry Wt. g | Efficiency of inoculants | Tolerance index % | Tio$_2$NPs concentrations | | |
|---|---|---|---|---|---|---|---|
| Treatments | | | | | Root | Shoot | T.F. |
| P. Control | NM | 4.6 | | | | | |
| | M | 11.7 | 154 | | | | |
| TiO$_2$NPs | NM | 4.9 | 6 | | 0.53d | 0.77a | 1.45 |
| | M | 13.6 | 195 | | 0.66b | 0.42c | 0.63 |
| NaCl 100mM | NM | 3.5 | -24 | 76 | - | - | |
| | M | 12.9 | 180 | 110 | - | - | |
| NaCl 200mM | NM | 1.9 | -59 | 38 | | | - |
| | M | 6.8 | 48 | 58 | - | - | |
| TiO$_2$NPs+100mMNaCl | NM | 3.2 | -37 | 65 | 0.47d | 0.79a | 1.68 |
| | M | 14.9 | 224 | 110 | 0.92a | 0.47a | 0.51 |
| TiO$_2$NPs+NaCl200mM | NM | 1.2 | -74 | 24 | 0.11e | 0.78a | 7.1 |
| | M | 10.6 | 130 | 78 | 0.91c | 0.48ab | 0.52 |
| LSD (P>0.05) | | 0.54 | | | 0.026 | 0.054 | |

**Efficiency of inoculants** = d.w treated- d.w control / d.w. control; **Tolerance index TI %** = d.w. Plant at salinity level ×100 / d.w. plant at 0.0 level of salinity; **NM**; Non-mycorrhizal plants and **M**; Mycorrhizal plants., **Translocation Factors (T F) = level of element in shoot / level of element in root;** LSD: At significant level (P>0.05).Sample symbols (a.a) mean non significant difference (a.b) mean significant difference.

decreased from 38% to 24% at 200mM of salinity. On the other hand, the results also clearly show that although the tolerance indexes of common bean plants were significantly decreased by increasing the salinity levels(P>0.05), plants treated with AMF still had high salinity tolerance than untreated one at all salinity levels. It was also quite clear that the presence of $TiO_2NPs$ with AMF resulted in increasing salinity tolerance of common bean plants at all salinity levels. It reached 110% at the salinity level of 100 mM NaCl.

From the results given in Table 4, it is also obvious that $TiO_2NPs$ accumulated in higher amount in shoots than roots of plants untreated with AMF either in presence or absence of the salinity stresses. On the other hand, the $TiO_2NPs$ accumulated in roots of plants treated with AMF than untreated one that growing either in saline or normal conditions. The results also showed that the translocation factors (TF) of $TiO_2NPs$ in plants treated with AMF were much lower than those untreated under salinity stress. This may indicate that the major role of mycorrhizal symbiosis in $TiO_2NPs$ translocation within plants.

The results given in Table 5 showed that increasing the level of salinity leads to decrease in the content of phosphorus (P) in the plant and that the presence of mycorrhiza reduces the harmful effects of salinity on common bean plants; where it was found that the content of phosphorus in plants treated with mycorrhiza remains always higher than untreated one. Moreover, the results showed that plants treated with ***Funneliformis mosseae*** and $TiO_2NPs$, exhibited the same ratio of phosphorus at the low level of salinity and at normal conditions, which ranged from 6.2 to0.65 mg/gm. The results of Table 5 confirmed that the potassium (K) content of the plant decreased by increasing salinity level. Plants treated with AMF showed higher potassium content than untreated one either in the presence or absence of salinity. The results also showed that plants co-inoculated with AMF and $TiO_2NPs$ displayed higher content of potassium than un-inoculated plants with value19.1 mg/gm at high level of salinity. The results given in Table 5 showed that the content of the nitrogen (N) in common bean plants is significantly lower under salinity stress in the absence of AMF (P>0.05). While in the presence of AMF, N content plants inoculated with AMF was not significantly affected by the presence of salinity, especially in the presence of $TiO_2NPs$ when compared with uninoculated plants.

**Table 5.  Effect of salinity levels on Phosphorus (P), potassium (K), Nitrogen (N) and protein concentrations, of and mycorrhizal, non-mycorrhizal plants supplemented with and without $TiO_2NPs$.**

| Periods(days) Treatments | | Mean of P content (mg/ g dry weight) | Mean of K content (mg/g dry weight) | Mean of N (mg/g dry weight plant) | Mean of Protein content (mg/ g dry weight plant |
|---|---|---|---|---|---|
| P control | NM | 4.8b | 10.9d | 5.3d | 6.2c |
| | M | 6.3a | 14.9c | 6.6b | 7.7a |
| $TiO_2NPs$ | NM | 5.2b | 3.5b | 5.3d | 6.2c |
| | M | 6.5a | 15.9a | 6.9bc | 8.3b |
| NaCl 100mM | NM | 0.6e | 2.8f | 0.8e | 1.1f |
| | M | 4.6b | 16.2e | 5.8e | 7.2b |
| NaCl200mM | NM | 0.45e | 4.6f | 0.7e | 1.0f |
| | M | 4.1b | 16.1e | 5.3e | 6.7b |
| $TiO_2NPs$+NaCl | NM | 1.9c | 7.7 c | 5.3d | 5.8d |
| 100mM | M | 6.2a | 18.4a | 6.4a | 8.9a |
| $TiO_2NPs$ +NaCl200mM | NM | 1.0d | 5.4d | 5.1e | 6.5e |
| | M | 5.2b | 19.1a | 6.2c | 7.9a |
| L.S.D(P>0.05) | | 0. 34 | 0. 54 | 0. 26 | 0. 54 |

**NM**; Non-mycorrhizal plants and **M**; Mycorrhizal plants. LSD: At significant level (P> 0.05). Sample symbols (a.a) mean non significant difference (a.b) mean significant difference.

The results also referred to that in the presence of $TiO_2NPs$, the N content in common bean plants under salinity was closely like those under normal condition. Similar results were obtained for the protein content in common bean plants where, the protein content of plants was significantly decreased under salinity conditions (Table 3). The results revealed that the presence of AMF amended with$TiO_2NPs$ alleviate the harmful effects of salinity on protein content of common bean plants.

The results in S1 Table showed that the plant content of sodium increased by increasing the salinity level in all treatments. Despite this increase, the total sodium content in the shoots of plants inoculated with AMF remained less than in the roots. It ranged from 12.3–18.2 in the shoots and to 28.4–59.0 in the root. The sodium content in the shoots of plants inoculated with AMF decreased in the presence of $TiO_2NPs$. The results also showed that the Translocation Factors (T.F) of sodium decreased in plants inoculated with AMF, especially in the presence of $TiO_2NPs$. The results confirmed the role of mycorrhiza in increasing plant tolerance for salinity, and this role is increased in the presence of $TiO_2NPs$. The results given in S1 Table illustrated that the calcium (Ca) content in the plant decreased by increasing the level of salinity. In addition, the concentration of Ca in the roots remains higher than that of the shoots and the values of T.F. in the presence of AMF higher than in the absence of AMF. A significant finding in the results was that the T.F. recorded 60–63% in the presence of AMF and $TiO_2NPs$ either in the presence or absence of salinity. The results of S1 Table showed the irregularity of the results for salinity, but magnesium (Mg) content was higher in plants inoculated with AMF than uninoculated one. The values of T.F. were higher in the presence of AMF and $TiO_2NPs$.

The results given in Table 6 showed that the activities of peroxidase and catalase enzymes in common bean plants increased either in the absence or the presence of saline conditions. The obtained results also revealed that activities of both enzymes were higher in plants inoculated with AMF than uninoculated one. The maximum activity recorded 0.57 and0.26 unit/mg of protein for peroxidase and catalase enzymes, respectively at 200mM salinity in plants

**Table 6. Effect of salinity levels on enzymes activity (Peroxidase, Catalase, acid & alkaline phosphates) of mycorrhizal, and non-mycorrhizal plants supplemented with and without $TiO_2NPs$.**

| Treatments | | Mean of Peroxidase activity (unit/ mg protein) | Mean of Catalase activity (Unit/mg protein) | Mean of acid Phosphates (mg p /mg protein/min (Unit/0.1 g fresh weight) | Mean of Alkaline phosphates (mg p /mg protein/min (Unit/0.1 g fresh weight) |
|---|---|---|---|---|---|
| P | NM | 0.10g | 0.06e | 12.30e | 9.00f |
| | M | 0.24b | 0.13c | 35.40c | 31.50b |
| $TiO_2NPs$ | NM | 0.11e | 0.08d | 12.30e | 10.20e |
| | M | 0.49a | 0.19a | 42.00a | 41.30a |
| NaCl 100mM | NM | 0.13e | 0.09c | 5.00g | 3.70g |
| | M | 0.54b | 0.21b | 32.27f | 30.50g |
| NaCl 200mM | NM | 0.14h | 0.1f | 4.55g | 2.99g |
| | M | 0.54c | 0.23b | 28.21f | 27.12g |
| $TiO_2$ NPs +100mM NaCl | NM | 0.15f | 0.11g | 10.30e | 10.00 |
| | M | 0.56d | 0.24c | 38.40b | 34.10c |
| $TiO_2NPs$+NaCl 200mM | NM | 0.15f | 0.13g | 09.0d | 10.10d |
| | M | 0.57d | 0.26c | 31.33d | 32.50c |
| L.S.D(P>0.05) | | 0.0093 | 0.014 | 0.14 | 0.14 |

NM; Non-mycorrhizal plants and M; Mycorrhizal plants. LSD: At significant level (P> 0.05).Sample symbols (a.a) mean non significant difference (a.b) mean significant difference.

inoculated with AMFandTiO$_2$NPs. On the other hand, the activities of phosphates enzymes (acid and alkaline) were significantly reduced (P>0.05) by raising salinity in uninoculated plants either in the absence or the presence of TiO$_2$NPs (Table 6). It is also obvious that the inoculation of common bean plants with the AMF resulted in stimulation of phosphatases enzymes, although the activities were slightly decreased in saline conditions. Moreover, the results also revealed that co-inoculation of common bean plants with AMF and TiO$_2$NPs-caused increases in acid and alkaline phosphatases compared with other treatments at all salinity levels.

The statistical analysis of data (Table 7) confirmed significant effects of salinity (P>0.05)on dry weight, Na, K, P, Ca accumulation, protein content and catalase and peroxidase activities. The results also showed that the combined effect of AMF and TiO$_2$NPs, AMF and salinity is similar to the combined effect of AMF, TiO$_2$NPs and salinity. On the other hand, the results in Table 7, confirmed that TiO$_2$NPsdoes not have significant effect on the growth of common bean plants under saline environment. The results indicated that the effect of TiO$_2$NPs on AMF was more positive than on the plants itself under saline conditions.

PCR amplification of the nest primer region using DNA extracted from AMF colonizing roots with various treatments recorded at 550bp (Fig 4). In addition, PCR amplification of the Chitin synthase primer region using DNA extracted from AMF colonizing roots with various treatments recorded at 580bp (Fig 5).The gene expression (molecular ratios intensity) analysis (S2 Table) for amplified nest and Chitin syntheses genes showed that AMF genes in the presence of TiO$_2$NPs had the most significant intensity and quantity compared to AMF genes in the absence of TiO$_2$NPs. On the hand, the intensity of AMF gene decreased significantly (P>0.05) under salinity stress. While the amendment of AMF with TiO$_2$NPs to significantly increased AMF genes intensity compared to AMF under salinity stress without TiO$_2$NPs. Thus, statistical molecular ratio analysis revealed that intensity of AMF genes was higher with TiO$_2$NPs. By this way, Arbuscular Mycorrhiza Relative Density (AMRD) was calculated for each treatment in roots of common bean plants. The analysis of variance (S2 Table) indicated that TiO$_2$NPs had a significant role on the relative density of mycorrhizal species in roots.

**Table 7. Variance (ANOVA) analysis of data for growth and physiological parameters and mycorrhiza levels of Phaseolus plants inoculated with *Funneliformis mosseae* and TiO$_2$NPs in the presence of salinity.**

| Treatments | M | M & S | N | M&N | N & S | S | M&N&S |
|---|---|---|---|---|---|---|---|
| Dry weight | ** | ** | ns | ** | * | ** | * |
| Sodium | * | ** | ns | ** | Ns | ** | ** |
| Calcium | * | * | ns | * | * | * | ** |
| Magnesium | * | * | * | * | ns | n | * |
| Potassium | * | * | ns | * | ns | * | * |
| Phosphorus | ** | * | * | * | ns | * | ** |
| Nitrogen | * | * | * | * | * | ns | ** |
| Protein content | ^ | * | * | ** | * | * | ** |
| Acid Phosphatases | ** | * | ns | ** | ns | ns | * |
| Alkaline Phosphatases | ** | * | ns | ** | ns | ns | * |
| Catalase | ns | ns | * | * | * | * | * |
| Peroxidase | ** | * | ns | ns | ns | * | * |

M = mycorrhiza N = TiO$_2$NPs S = salinity

**Highly significant, P ≤0.01

* significant, P ≤ 0.01 ns = Non-significant.

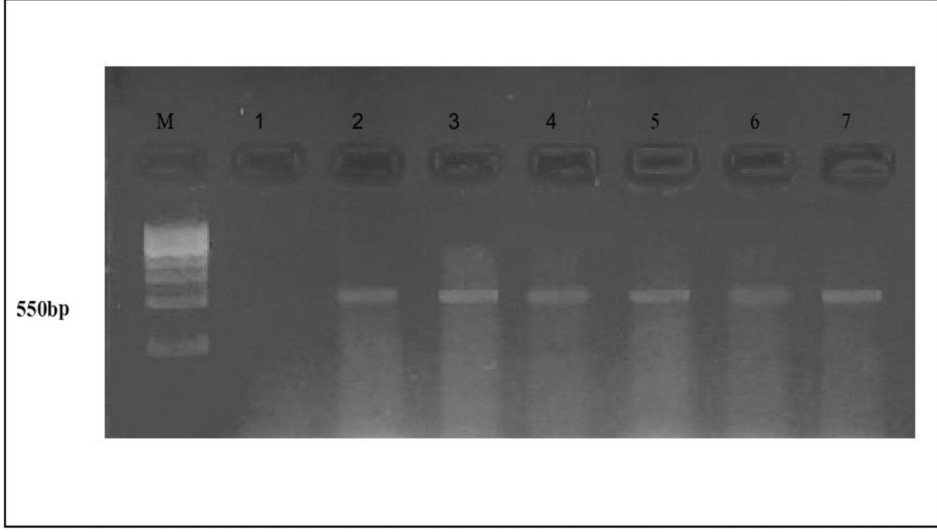

**Fig 4. PCR amplification of the nest primer region using DNA extracted from AMF colonizing roots with various treatments.** (M) DNA Marker; Lane (1) control (uninoculated with AMF), Lane (2)control (inoculated with AMF*)*; Lane (3), AMF + TiO$_2$NPs; Lane (4), AMF+ NaCl100mM; Lane (5), AMF+NaCl100mM+TiO$_2$NPs; Lane (6) AMF + NaCl 200mM; Lane (7) AMF+ NaCl200mM+TiO$_2$NPs.

*Funneliformis mosseae* nest and ChS relative density increased in roots when inoculated with TiO$_2$NPs (Figs 6 and 7). Also, TiO$_2$NPs increased the relative density of AMF under salinity stress compared to AMF under salinity stress without TiO$_2$NPs.The analysis indicated that relative density of nest genes of AMF increased 30%after inoculation with TiO$_2$NPs. The analysis of variance indicated that the AMF nest genes with TiO$_2$NPs had a significant effect on the relative density of AMF in roots compared with single inoculants of AMF under the presence or

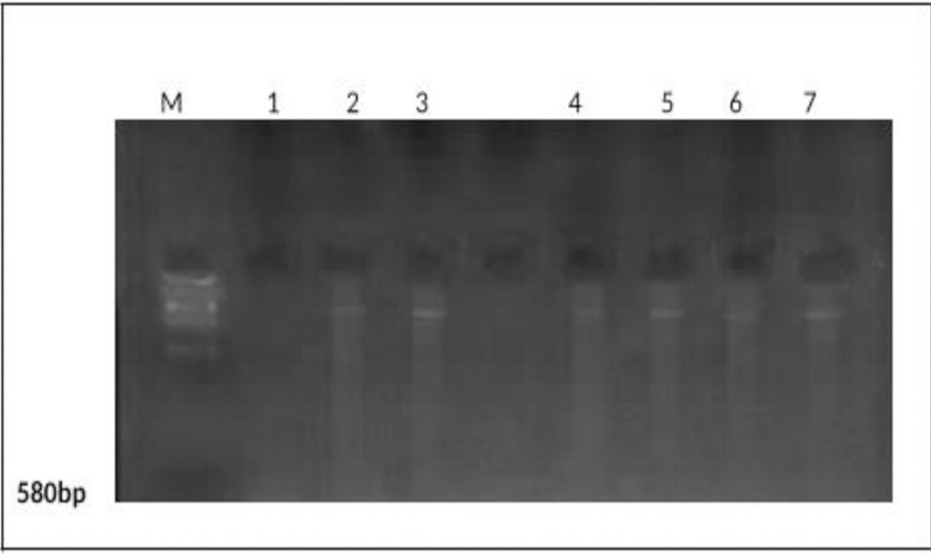

**Fig 5. PCR amplification of the Chitin synthase primer region using DNA extracted from AMF colonizing roots with various treatments.** (M) DNA Marker; Lane (1) control (uninoculated with AMF), Lane (2) control (inoculated with AMF*)*; Lane (3), AMF + TiO$_2$NPs; Lane (4), AMF+ NaCl100mM; Lane (5), AMF+NaCl100mM+TiO$_2$NPs; Lane (6) AMF+ NaCl 200mMl; Lane (7) AMF+ NaCl 200mM+TiO$_2$NPs.

.

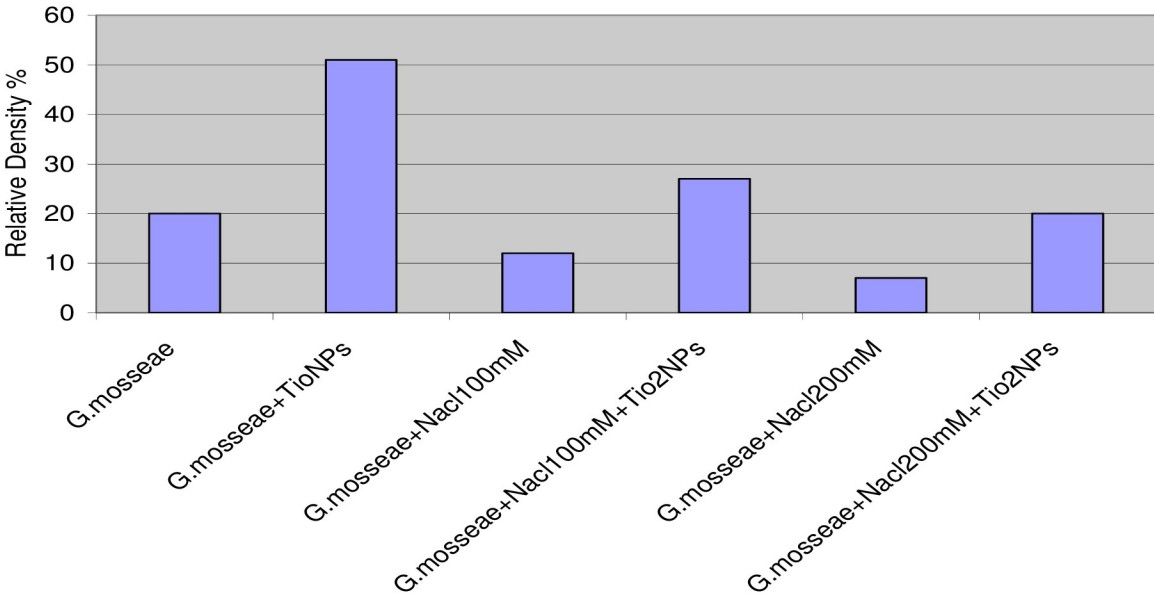

Fig 6. Molecular relative density of nest gene intensity and quantity fold under different treatments: Single AMF, paired inoculation of AMF with TiO$_2$NPs, AMF under different salinity stress and AMF with TiO$_2$NPs under different salinity stress.

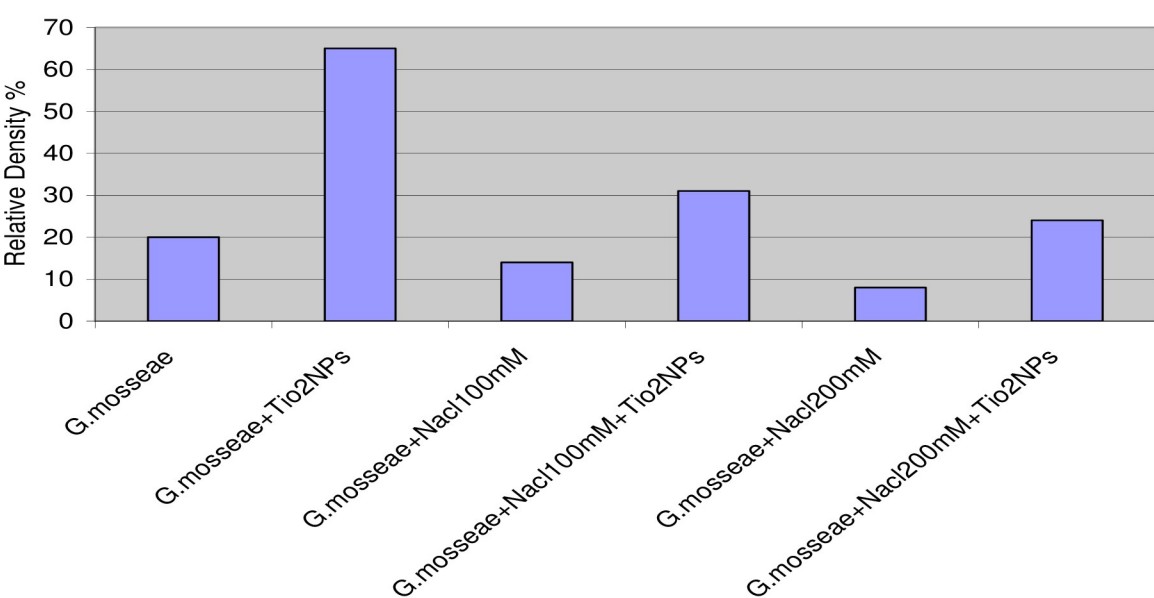

Fig 7. Molecular relative density of gene intensity and quantity fold under different treatments: Single AMF, paired inoculation of AMF with TiO$_2$NPs, AMF under different salinity stress and AMF with TiO$_2$NPs under different salinity stress.

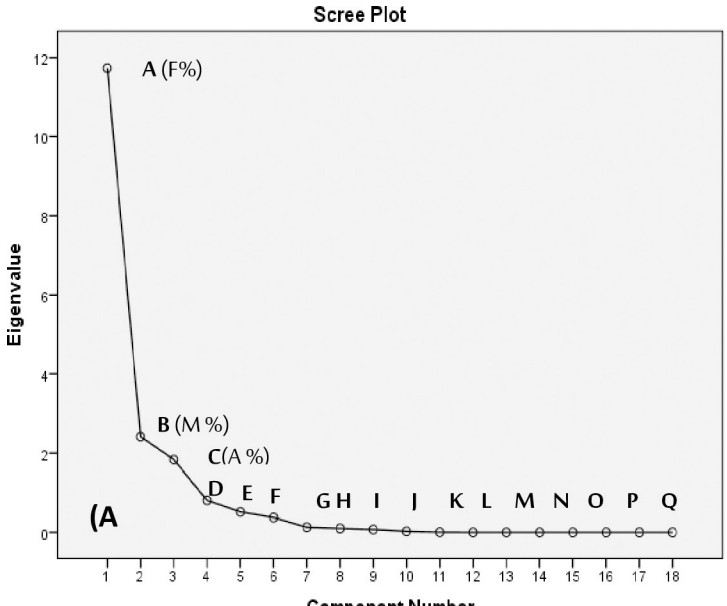
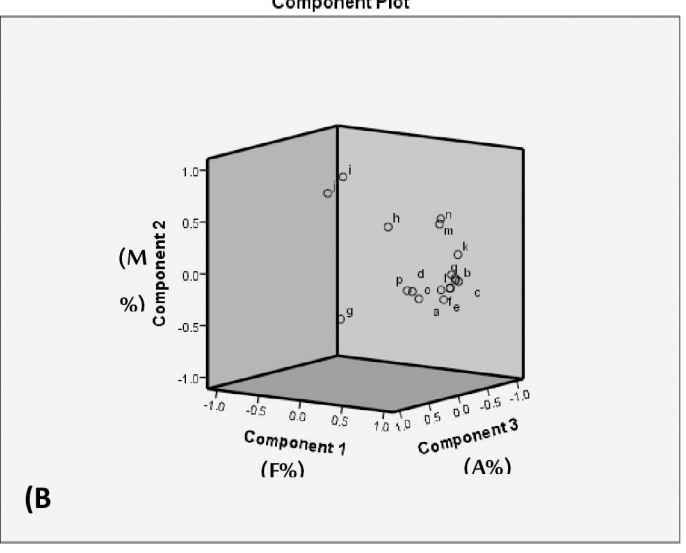

**Fig 8.** The scree plot of eigenvalue (A) and loadings plot (B) for the component number of the measured parameters for common bean plants under salinity stress in response to treatments with AMFandTiO₂NPs: **A:**F% Frequency of mycorrhizal root segments, **B:**M% intensity of mycorrhizal colonization, **C:**A% arbuscule frequency in root systems, **D:**(MD) mycorrhizal dependency, **E:**Dry Wt. (g), **F:**Efficiency of inoculants, **G:**Tolerance index %, **H:**Tio₂NPs concentration in root, **I:**Tio₂NPs concentration in shoot, **J:**(T.F) Translocation Factors, **K:**(P) content, **L:**(K) content, **M:**(N) content, **N:**Protein content, **O:**Peroxidase activity, **P:**Catalase activity, **Q:** Acid Phosphates and **R:** Alkaline phosphates.

the absence of salinity stress about 30% and 40%, respectively (Fig 6). In addition, the presence of TiO₂NPs increased the relative density of the chitin synthase gene of AMF under salinity stress compared to single inoculants of AMF about (50%), (Fig 7). Thus, TiO₂NPs play a major role in the significant increase of AMF activity, intensity and relative density.

Principal component analysis (PCA) was performed on measured parameters of common bean plants under salinity stress in response to treatments with AMF and TiO₂NPs (Fig 8A and 8B). Interpretation of the principal components (PCs) was aided by inspection of the factor-loading matrix extracted from a Varimax rotation with Kaiser Normalization of main components to identify factors responsible for the grouping of the dataset. As shown from scree plot graph of eigenvalues and loadings plot (Fig 8A and 8B), the first three components; F %, M % and A % have higher eigenvalues than other measured components. Therefore, the two components were extracted describing approximately 88.81%. Accordingly, principal component analysis gave evidence to the importance of parameters *i.e.* M (%) and A (%), especially F (%), as it was considered a good and more reliable indicator for development of AMF in the presence and absence of TiO₂NPs under salinity stress.

## Discussion

Mycorrhizal symbiosis is an opener compound in assisting plants to overcome counteractive environmental situations. In this study, the overall growth and physiological parameters of common bean plants grown under two levels of salinity increased upon association with AMF. The results also indicated that application of TiO₂NPs can improve the growth of inoculated and uninoculated plants with AMF at the two levels of salinity but in lesser extent in uninoculated plants. These findings suggested that application of AMF combined with TiO₂NPs may help in overcoming the detrimental growth effects of salt stress induced by sequential

irrigation with salt solutions at100 and200 mM NaCl. As expected and documented in the literature, AMF has been demonstrated to induce salinity tolerance in various plant species [48,49].

In the present study, mycorrhizal colonization of common bean roots were significantly reduced under salinity levels but plants inoculated with AMFandTiO$_2$NPs still had higher mycorrhizal colonization. In contrast, [50] it was demonstrated that the inhibitory effect of TiO$_2$NPs on arbuscular mycorrhizal symbiosis in roots of rice plant at medium to higher levels of application ($>$ 2%). Notably, it was observed that TiO$_2$ NPs had a negative effect on the abundance of some prokaryotic and increased the abundance of other prokaryotic and some fungal organisms. Some of these fungal organisms belonged to the phylum Ascomycota [51]. However, concerns have been raised that some nanoparticles may negatively affect crops, soil microbial communities and their effects on the diversity and community composition of soil microorganisms including beneficial microbes such as arbuscular mycorrhizal fungi and this depends on the concentration of TiO$_2$NPs that used [27]. Overall, it was proved that prokaryotes are more sensitive than fungi to the TiO$_2$NPs treatments, also No negative effects on arbuscular mycorrhizal root colonization were detected, and no evidence for a dose-response relationship between wheat performance and TiO$_2$NPs concentration was found [27].Previous literature suggested that high concentrations of silver nanoparticles can deteriorate the mutual interaction between plants and AM fungi and negatively influence the rhizospherics oil P cycling, both of which go against plant growth and soil fertility. The effects of silver nanoparticles on plants and soil microbial communities have been widely documented [52,53].

One of the most interesting results in Table 3, is the values of mycorrhiza development at the end of the experiment (90 days) was higher than that at 45 days in the presence of TiO$_2$NPs under salinity conditions. In this connection [54], it was found that the inhibitory effect of TiO$_2$NPs on AMF colonization in plant roots was more pronounced during early stages of its application and found to recover during the late stages of application. For this reason, it could be explained that AMF activities were not affected by the inhibitory effect of TiO$_2$NPs at the end of the experiment. One explanation based on the high mobility of TiO$_2$NPs through the porous substratum *i.e.* soil. Another one is the production of more mycorrhizal roots in the absence of a high concentration of TiO$_2$NPs for a higher scoring of mycorrhizal variables at late stages of application. The amazing results in this study, which supported our previous results, is activity of mycorrhizal measurements (F%, % M, % A) which were increased in the presence of TiO$_2$NPs, especially for arbuscule activity (% A) which was three times that of the absence of TiO$_2$NPs. Since arbuscule meant for the transference of nutrients between fungus and the plant, the increasing activities of this structure are likely to affect the functional symbiosis within the host roots leading to increase in growth and tolerance of the plant to salinity. Evidence from our data, indicates that mycorrhizal dependencies for plant dry mass increased by raising salinity. The increase varied from 369% to 465% in the absence and presence of TiO$_2$NPs, respectively at lower salinity level. While at higher salinity level, dry mass was increased from 358% to 883% in the absence and presence of TiO$_2$NPs, respectively. The beneficial effects of mycorrhiza on the growth under saline conditions have been studied in various plants [55,56].

In the present investigation the tolerance index common bean plants inoculated with AMF was higher than that uninoculated one at all levels of salinity. The tolerance index of salinity in plants inoculated with AMF increased in the presence of TiO$_2$NPs. Tolerance index of salinity in plants inoculated with AMF and TiO$_2$NPs were nearly 2.0 and 3.0 fold higher than that of uninoculated plants, respectively at the two salinity levels. In addition, one of the interesting findings in this study was that the rate of improvement of plants growth under saline conditions in the presence of AMF and TiO$_2$NPs was better than in AMF only. While, TiO$_2$NPs

alone does not have a significant effect on plant growth in this conditions. The improvement in the growth of common bean plants inoculated with AMF in the saline environment was due to the influence of $TiO_2NPs$ on the efficacy of AMF and not on the plant itself.

Nutrient imbalances in plants may create from the effect of salinity on nutrient availability, competitive uptake, transport or partitioning within the plant, or may be caused by physiological inactivation of a given nutrient. In the present work, the ions deficiency displayed by salinity stress, particularly by NaCl uptake indicated a nutritional imbalance. It has been generally accepted that Am fungi would elevate nutrient uptake by infected plants under salinity conditions [57]. In the results obtained herein, it is interesting to note that common bean plants inoculated with AMF contained significant levels N, P and K, particularly in the presence of $TiO_2NPs$. Based on these results and available literature, the greater salt tolerance of common bean plants inoculated with AMF may be the result of the plant nutrition improvement under salinity stress. Previous researches have emphasized the potency of AMF in minimizing the negative influences of salinity on various plants [58].

Surprisingly, while mycorrhizal plants in the present study gathered more Na+ in their roots with raising salinity, shoots of these plants had lower Na+ content and exhibited limited collection with increasing salinity particularly in the presence of $TiO_2NPs$, and in absence and presence of salinity. These results are consistent with the previous work [59] and proposed that AMF may conserve the shoot system particularly leaves from Na+ toxicity either by accumulating it in root thereby dilatory its translocation onto shoot system of infected plants or by adjusting $Na^+$ uptake from the soil. On the other hand, the results obtained herein demonstrated that $K^+/Na^+$ ratio was higher in plants inoculated with AMF, particularly in presence of $TiO_2NPs$, under salinity levels. In this context, the higher ratios of $K^+/Na^+$ was proved to be one of the key determinants of plant salt tolerance [60].

The results of this study specified that AMF symbiosis created by a salt-tolerant fungus can significantly raise the dry weight, P, K, N, protein content, Mg, catalase, peroxidase, acid and alkaline phosphatases activities of common bean plants under saline stress conditions particularly in the presence of $TiO_2NPs$.These results proved that the incipient role of AMF in increasing salinity tolerance of common bean plants up to 200 mM of NaCl. In addition, the results demonstrated that monitoring of ion uptake by roots and transport into leaves and compartmentalization of ions at the cellular and whole plant levels are the ultimate efficient action of AMF for acclimation of common bean plants to salinity stress [61,62]. These delineations have been enhanced efficiently by co-inoculating AMF with $TiO_2NPs$. However, little information is currently available about potentiality of $TiO_2NPs$ as co-activator to AMF under stress. Studies on AMF and$TiO_2NPs$interactions are still in their infancy; more research is needed to better understanding of the interactions between AMF and nanoparticle materials for the development of sustainable management of soil salinity and other stresses.

The previous research about $TiO_2NPs$ potency to be applied as a "stress-ameliorative" or "chemical-priming" agent in plants or in the rhizosphere were little and scattered. It was shown that the growth intensifying influences of low-concentration of $TiO_2NPs$ as well as its ameliorative potential in dealing with soil salinity [63]. The influences related to $TiO_2NPs$ ability to boost photosynthesis and antioxidant defense, which help plants intensify their growth. A recent [64] demonstrated that the ameliorative effects of $TiO_2NPs$, against cold stress on chickpea plants had been related to the possible responsive components that were involved in signal transduction, defense, metabolism and regulatory processes. Apparently, these components supply positive support in enhancing the tolerance of chickpea to cold stress. In consistency with these results, our results reveal that Phosphorus, nitrogen, protein, magnesium and catalase were significantly increased in common bean plants in the presence of $TiO_2NPs$ under salinity stress. In this connection, [65] it was proved that the presence of $TiO_2NPs$ could

also ameliorate plant nitrogen status. Furthermore, it was found that application of$TiO_2$NPs boosted the enzymatic antioxidant defense, which helped plants realize better conduct of raised ROS levels output under salinity [17].

Apart from the results of salinity stress, careful examination of our results revealed that plant inoculated with AMF showed higher significant increases in all parameters measured in the presence of $TiO_2$NPs, while the presence of $TiO_2$NPs alone in common bean plants did not exhibit significant effect in most measurement. From the results obtained herein, the plant growth response was assessed by whole plant biomass and nutrient analysis, while AMF activity responses were determined using DNA-based methods or biomass measurements as needed. The initial application of the PCR technique has been very successful [42]. In this study, DNA extraction and the PCR amplification from roots showed an expected fragment size for the nest and ChS primers. So, we concluded that AMF is capable of colonizing common bean roots [66,67]. From the results obtained, the molecular ratio and relative density were two folds in AMF with $TiO_2$NPs compared with the single inoculation of AMF or under the presence of salinity stress in roots [11]. In this connection, the results of this study showed that the different treatments of paired inoculation with AMF could have different effects on AMF colonization. Whereas, the relative density and the intensity of gene expression in AMF in the presence of $TiO_2$NPs were higher than AMF alone or in the presence of salinity stress [68]. So, $TiO_2$NPs have a stimulatory effect on the establishment of the AMF symbiosis [69,70].Thus, the intensity detection of polymerase chain reaction (PCR) products in this study made by a reaction of the molecule that reports an increase in the amount of DNA with a proportional increase in intensity [40].Where specific primers were applied to quantify mycorrhizal abundance and activity in the plant roots affected by $TiO_2$NPs in the presence and absence of salinity treatments employing quantitative PCR.

## Conclusions

In the current study, the inoculation of AMF along with$TiO_2$NPs showed to be highly effective in improving salinity tolerance compared to single inoculation with AMF alone. In addition, the presented study revealed that PCR method considered as a quantitative detection system for evaluation the occupancy and the activity by symbiotic and mutalistic relationship between AMF and common bean plants. Accordingly, it is of interest to suggest that the positive effect of $TiO_2$NPs was confined to its effect on AMF not on common bean plant itself. Therefore, a better understanding of the interactions between $TiO_2$NPs and AMF activities within plants responses, including their uptake transport, internalization, and activity, could revolutionize crop production through increased disease resistance, nutrient utilization, and crop yield.

## Supporting information

**S1 Table. Effect of salinity levels on sodium (Na), calcium (Ca), magnesium (Mg) and translocation.**
(DOCX)

**S2 Table. Amplified Nest and chitin synthase quantity (intensity & relative density) of mycorrhiza levels of Phaseolus plants inoculated with *Funneliformis mosseae* and $TiO_2$NPs in the presence of salinity.**
(DOCX)

**S1 File. Brief methodologies protocol.** Brief methodologies protocol for some measurements.
(PDF)

**S1 Raw images.**
(PDF)

## Acknowledgments

The authors would like to express them heartfelt thanks to both Prof. Dr. Gamal Enan, Professor of Microbiology, Dean of Faculty of Science, Zagazig University, Egypt and Prof.Dr. Gamal Rabie, Professor of Microbiology, Faculty of Science, Zagazig University, Egypt for their guides, instructions and revising assistance in this work. We are also indebted to King Saud University, Riyadh, Saudi Arabia for facilities.

## Author Contributions

**Conceptualization:** Nashwa El-Gazzar.

**Data curation:** Nashwa El-Gazzar.

**Formal analysis:** Nashwa El-Gazzar.

**Funding acquisition:** Khalid Almaary.

**Investigation:** Nashwa El-Gazzar, Ahmed Ismail, Giancarlo Polizzi.

**Methodology:** Nashwa El-Gazzar.

**Project administration:** Nashwa El-Gazzar, Khalid Almaary, Giancarlo Polizzi.

**Software:** Nashwa El-Gazzar.

**Supervision:** Khalid Almaary, Giancarlo Polizzi.

**Visualization:** Ahmed Ismail, Giancarlo Polizzi.

**Writing – original draft:** Nashwa El-Gazzar.

**Writing – review & editing:** Nashwa El-Gazzar, Ahmed Ismail.

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
