## [Decision Letter · Decision Letter 0]

4 May 2020

PONE-D-19-32967

Influence of Arbuscular mycorrhizal fungi (AMF) enhanced with titanium dioxide nanoparticles on common bean (Phaseolus vulgaris L.) plants under salinity stress

PLOS ONE

Dear Dr. El-Gazzar,

Thank you for submitting your manuscript to PLOS ONE. After careful consideration, we feel that it has merit but does not fully meet PLOS ONE’s publication criteria as it currently stands. Therefore, we invite you to submit a revised version of the manuscript that addresses the points raised during the review process.

Please revise your manuscript according to the comments from both Reviewers below.

We would appreciate receiving your revised manuscript by Jun 18 2020 11:59PM. To enhance the reproducibility of your results, we recommend that if applicable you deposit your laboratory protocols in protocols.io, where a protocol can be assigned its own identifier (DOI) such that it can be cited independently in the future. For instructions see: http://journals.plos.org/plosone/s/submission-guidelines#loc-laboratory-protocols

We look forward to receiving your revised manuscript.

Kind regards,

Ying Ma, Ph.D.

Academic Editor

PLOS ONE

Reviewers' comments:

Reviewer's Responses to Questions

**Comments to the Author**

1. Is the manuscript technically sound, and do the data support the conclusions?

Reviewer #1: Yes

Reviewer #2: Yes

2. Has the statistical analysis been performed appropriately and rigorously? 

Reviewer #1: Yes

Reviewer #2: Yes

3. Have the authors made all data underlying the findings in their manuscript fully available?

Reviewer #1: Yes

Reviewer #2: Yes

4. Is the manuscript presented in an intelligible fashion and written in standard English?

Reviewer #1: Yes

Reviewer #2: Yes

5. Review Comments to the Author

Reviewer #1: Dear editor,

I would like to thank the opportunity to revise this paper. This paper has an interesting overall goal to understand the influence of nanotechnology on arbuscular mycorrhizal fungi (amf) that in turn influence Phaseolus vulgaris plants. It is a nice paper and should be accepted for publication in PLOS ONE. I have some consideration, comments and corrections (I follow lines in pdf MS, line 1. Title of article)

To change the title by e.g., “Influence of Funneliformis mosseae enhanced with titanium dioxide nanoparticles on Phaseolus vulgaris L under salinity stress”, since nowadays Glomus mosseae is classified as Funneliformis mosseae and it also can increase the interest of the potential readers;

Delete abbreviation of short expression, use full words. Abbreviations typically decrease the number of readers because is interdisciplinary;

Some keywords do not reflect the content of the manuscript. I suggest using the following keywords:

Glomeromycota instead Microbiology and Mycology;

Funneliformis mosseae instead Glomus mosseae

Common bean roots instead Plant science

Activity of arbuscular mycorrhizal fungi instead life science and environmental sciences

Nanotechnology

Salinity stress

In the introduction, please make it more clearly what is the novelty (scientific/technical) of your work, approach, and results compared to already well-known facts. Most facts and trends given in the text of this article are much too vague. All trends, facts, and results must be supported by precise terms and data/numbers;

Please, to provide clear hypotheses based in more than two scientific statements;

L53-54 I can’t see the linkage to the above paragraphs;

L66 Funneliformis mosseae instead Glomus mosseae;

L69 The description of the “source of mycorrhiza, pot and potting mixture” section is not clear. I suggest it be made clearer to facilitate the evaluation of the MS;

L82 Use “chemical analysis” instead “biochemical analysis;

L83 soil pH?

L85-86 The pots and soil were sterilized?

L107 How many blocks?

L108-113 The description of the treatments is not clear. I suggest to use a table to facilitate the reader to see the differences among the studied treatments;

L108, 109, 110, 111, 112, 113 Funneliformis mosseae instead Glomus mosseae;

L118 To many dissimilar variables into the same subsection. I suggest split it in four subsections: 1) AMF (e.g., Colonization and mycorrhizal dependency); 2) Plant biomass and protein; 3) A&A P activity; and 4) Plant macronutrient contents. Also, the authors must describe all methods clearer;

L21, 118 Use the word “biomass” instead “mass” or even “matter” to describe plant/shoot/root dry biomass;

L133 Since AMF are classified into the Phylum Glomeromycota, I suggest using only the word “spores” instead “Chlamydospores”;

L191 I expected to see a principal component analysis or even a non-metric multidimensional scaling to describe the similarities/dissimilarities among the treatments considering all matrix;

The presentation of the results can be improved using multivariate analysis.

To many tables. I suggest presenting only significative results. The results without significative differences must be added as an electronic supplementary material;

The discussion needs to be improved by explaining further the results and the mechanisms underlying salinity, the use of nanoparticles and the growth of inoculated plants with AMF;

Reviewer #2: The use of nanotechnology for improving the soil-plant relationship is an emerging area of research with wide academic and industrial scope. This paper tried to investigate the role of titanium dioxide nanoparticles in improving the AMF assisted and plant-soil interactions. The work seems to be carried out with care and the findings reported by the authors have importance in improving the crop productivity in saline soils. Before its publication, authors are required to improve the manuscript by following suggestions. I recommend moderate revision to the manuscript.

1. The title should be revised by adding “(TiO2NPs)” particularly after titanium dioxide.

2. Line 21: Add “, respectively” after salinity level.

3. The introduction is nicely written and the problem is discussed well.

4. The authors should add some reports based on the use of NPs for enhancing the soil-plant interactions to strengthen the proposed hypothesis in the introduction section.

5. Line 61: Thus, ....., the line should be started from a new paragraph. It will bring clarity between the novelty and objectives of the study.

6. Line 74: Describe why the authors selected the particular local strain of mycorrhiza with its industrial possibilities.

7. Line 75: The authors should provide a brief methodology for this section. The data may be provided as supporting information files to avoid text plagiarism, if.

8. Line 80: Provide geocoordinates of Sharkia province.

9. Most of the methodologies are just mentioned in short sentences which are not recommended. Provide brief protocols for them.

10. Line 85: Why you have used an abbreviated form of Dept. Agric. Res. Centre, Giza, Egypt?

11. Line 92: add space between TiO2NPswere.

12. Line 98: Direct use of numerical value in the starting of paragraph and sentence? Why?

13. Check for potential typing errors “breaks, spaces, dots, etc., throughout the manuscript.

14. Line 191: Provide names of software used in this study.

15. Line 197: Better to provide the information of the FTIR spectrum (9 peaks) as a single table data.

16. Some asterisks given in the table data are not defined in their footer captions.

17. Use uniform font and text formatting in the manuscript.

18. The level of statistical significance should be provided in the manuscript wherever it is mentioned.

19. Have you observed some negative effects of low NP treatment and high salt concentration? Describe in discussion section accordingly with supportive literature.

20. Many references are outdated. Authors are required to replace them with the latest one including https://doi.org/10.1016/j.bcab.2019.101463.

21. Figs. resolution is too low and difficult to understand.

6. PLOS authors have the option to publish the peer review history of their article (what does this mean?). If published, this will include your full peer review and any attached files.

Reviewer #1: Yes: Tancredo Souza

Reviewer #2: No

---

## [Author Response · Author response to Decision Letter 0]

7 Jun 2020

All the remarks of the reviewers are considered and corrections were given and marked by yellow colour. We declare that we have not ethical or legal restrictions for showing our data. We also do not have some things to keep in repositories and no accession numbers for organisms in Gene Bank are provided. All the original uncropped and unadjusted blot/ gel images reported in S1 raw images pdf file. 

1-The revised manuscript followed the PLOS ONE style requirements and the required files followed the naming provided in your last message as Instructions for authors of PLOS ONE. 

2- We certify that PLOS ONE only allows data to be available upon request. We also have not ethical restrictions on sharing our data for publicity in media or something else. 

I will be very happy if this revised manuscript could be published in PLOS ONE as an original contribution

Responses to reviewers

I would like to thank the reviewers

Reviewer 1

To change the title by e.g., “Influence of Funneliformis mosseae enhanced with titanium dioxide nanoparticles on Phaseolus vulgaris L under salinity stress”, since nowadays Glomus mosseae is classified as Funneliformis mosseae and it also can increase the interest of the potential readers;

I thank the reviewer by providing us with the recent update about the taxonomic issues of mycorrhiza. The correct and the name given by the reviewer is provided in the text. 

Delete abbreviation of short expression, use full words. Abbreviations typically decrease the number of readers because is interdisciplinary;

We followed the instruction for authors as can as possible. The acronyms are deleted with exception of some mandatory one. 

Some keywords do not reflect the content of the manuscript. I suggest using the following keywords:

Glomeromycota instead Microbiology and Mycology;

Funneliformis mosseae instead Glomus mosseae

Common bean roots instead Plant science

Activity of arbuscular mycorrhizal fungi instead life science and environmental sciences

Nanotechnology

Salinity stress

The keywords suggested are given and really are more understanding.

In the introduction, please make it more clearly what is the novelty (scientific/technical) of your work, approach, and results compared to already well-known facts. Most facts and trends given in the text of this article are much too vague. All trends, facts, and results must be supported by precise terms and data/numbers;

This point is very important and we thank the reviewer. The introduction was written again taking in account the last published perspectives in this field and the reviewer’s comments become more understanding.

Please, to provide clear hypotheses based in more than two scientific statements;

L53-54 I can’t see the linkage to the above paragraphs;

Introduction now is a bulk unit containing a sequence of thoughts and shows the need to do such work.

L66 Funneliformis mosseae instead Glomus mosseae;

It was changed within the whole manuscript

L69 The description of the “source of mycorrhiza, pot and potting mixture” section is not clear. I suggest it be made clearer to facilitate the evaluation of the MS;

The providing of the stock soil containing AFM was showed as it was part of our previous work. The stimulation of AFM growth and preparation of inocula are given in the text.

L82 Use “chemical analysis” instead “biochemical analysis;

This note is considered and corrected in the text

L83.soilpH?

pH at (7.9) and present within the manuscript

L85-86 The pots and soil were sterilized?

The soil was sterilized by autoclave and its details are given in the text. The pots were sterilized by their keeping in UV chamber for 12 h

L107 How many blocks?

6 replicates were allowed for each treatment. About 12 treatments as given in Table (1) were carried out

 L108-113 The description of the treatments is not clear. I suggest to use a table to facilitate the reader to see the differences among the studied treatments;

I thank the reviewer for such comment. Treatments are given in Table (1). 6 replicates from each treatment were carried out. 

L108, 109, 110, 111, 112, 113 Funneliformis mosseae instead Glomus mosseae;

It was corrected in the whole manuscript.

L118 To many dissimilar variables into the same subsection. I suggest split it in four subsections: 1) AMF (e.g., Colonization and mycorrhizal dependency); 2) Plant biomass and protein; 3) A&A P activity; and 4) Plant macronutrient contents. Also, the authors must describe all methods clearer;

This is good comment and subsections were made according to the reviewer comment and such brain storm really improved the manuscript. 

L21, 118 Use the word “biomass” instead “mass” or even “matter” to describe plant/shoot/root dry biomass;

It was changed in the manuscript

L133 Since AMF are classified into the Phylum Glomeromycota, I suggest using only the word “spores” instead “Chlamydospores”;

This is corrected in the whole manuscript and pointed by yellow colour

L191 I expected to see a principal component analysis or even a non-metric multidimensional scaling to describe the similarities/dissimilarities among the treatments considering all matrix;

I thank the reviewer for such comment. The principal component analysis and a non-metric multidimensional scaling already made among the treatments considering all matrix with total Variance and Component Matrixa and inserted as (Figure 8 a, b) in Figure file also explained in result section (last paragraph) by yellow colour. Principal component analysis (PCA) was performed on measured parameters of common bean plants under salinity stress in response to treatments with AMF and TiO2NPs (Fig. 8A and B). Interpretation of the principal components (PCs) was aided by inspection of the factor-loading matrix extracted from a Varimax rotation with Kaiser Normalization of main components to identify factors responsible for the grouping of the dataset. As shown from scree plot graph of eigenvalues and loadings plot (Fig. 8A and B), the first three components; F %, M % and A % have higher eigenvalues than other measured components. Therefore, the two components were extracted describing approximately 88.81%. Accordingly, principal component analysis gave evidence to the importance of parameters i.e.M (%) and A (%), especially F (%), as it was considered a good and more reliable indicator for development of AMF in the presence and absence of TiO2NPs under salinity stress. 

The presentation of the results can be improved using multivariate analysis.

Variance ANOVA analysis was done and showed the variance and significance or not. Results were taken as means and there are showed as a, b, c, d in tables; each with certain weight; LSD: At significant level (P> 0.05).Sample symbols (a.a) mean non significant difference (a.b) mean significant difference

To many tables. I suggest presenting only significance results. The results without significance differences must be added as an electronic supplementary material;

Both Table 4 and 7 as labeled in un revised manuscript , that without mandatory significance were changed to supplementary material Supplementary Table 1. Effect of salinity levels on sodium (Na), calcium (Ca), magnesium (Mg) and translocation. 

 Supplementary Table 2 . Amplified Nest and chitin synthase Quantity (Intensity & Relative density) of mycorrhiza levels of Phaseolus plants inoculated with Funneliformis mosseae and TiO2NPs in the presence of salinity. 

The discussion needs to be improved by explaining further the results and the mechanisms underlying salinity, the use of nanoparticles and the growth of inoculated plants with AMF;

These concepts and explanations are given in the text.

Reviewer #2: 

1. The title should be revised by adding “(TiO2NPs)” particularly after titanium dioxide.

It is done and mentioned in the title

2. Line 21: Add “, respectively” after salinity level.

It is done and mentioned in the title

3. The introduction is nicely written and the problem is discussed well.

Thank you very much for this point. Introduction is rewritten and discussed again.

4. The authors should add some reports based on the use of NPs for enhancing the soil-plant interactions to strengthen the proposed hypothesis in the introduction section.

Thank you very much for this point. Introduction was rewritten again taking in account the reviewer remarks.

5. Line 61: Thus, ....., the line should be started from a new paragraph. It will bring clarity between the novelty and objectives of the study.

This error is corrected

6. Line 74: Describe why the authors selected the particular local strain of mycorrhiza with its industrial possibilities.

This was our isolate and was used by our Lab in a previous work (Rabie et al., 2005).

7. Line 75: The authors should provide a brief methodology for this section. The data may be provided as supporting information files to avoid text plagiarism, if.

This is given in details and become more understanding.

8. Line 80: Provide geocoordinates of Sharkia province.

The geocoordinates of Sharkia province is provided within the manuscript. It follows Egypt.

9. Most of the methodologies are just mentioned in short sentences which are not recommended. Provide brief protocols for them.

Thank you for this comment. However, certain length of manuscript is allowed for each manuscript. Some of them are given as supplementary file.

10. Line 85: Why you have used an abbreviated form of Dept. Agric. Res. Centre, Giza, Egypt?

This abbreviation is changed to the whole word

11. Line 92: add space between TiO2NPswere.

It is done.

12. Line 98: Direct use of numerical value in the starting of paragraph and sentence? Why?

Such mistakes are corrected as each sentence contained subject +verb+ the rest of the sentence. Sorry.

13. Check for potential typing errors “breaks, spaces, dots, etc., throughout the manuscript.

The grammar and the whole manuscript are reviewed again by Prof. Dr. G. Enan.

14. Line 191: Provide names of software used in this study.

The software is added in the manuscript (WASP software version 2.0).

15. Line 197: Better to provide the information of the FTIR spectrum (9 peaks) as a single table data.

The information of the FTIR spectrum (9 peaks) is already presented in a single table 2.

16. Some asterisks given in the table data are not defined in their footer captions.

These points are very important; the corrections were made and explained. 

17. Use uniform font and text formatting in the manuscript.

It was done within the whole manuscript.

18. The level of statistical significance should be provided in the manuscript wherever it is mentioned.

Thank you very much for this point. The level of statistical significance is showed in the manuscript.

19. Have you observed some negative effects of low NP treatment and high salt concentration? Describe in discussion section accordingly with supportive literature.

This is done in the manuscript and some paragraphs are inserted in the discussion section and pointed by green colour.

20. Many references are outdated. Authors are required to replace them with the latest one including https://doi.org/10.1016/j.bcab.2019.101463.

As update of references were made and recent ones were added. 

21. Figs. resolution is too low and difficult to understand.

The quality of the figures improved.

---

## [Decision Letter · Decision Letter 1]

15 Jun 2020

Influence of Funneliformis mosseae enhanced with titanium dioxide nanoparticles (TiO2NPs) on Phaseolus vulgaris L. under salinity stress

PONE-D-19-32967R1

Dear Dr. El-Gazzar,

We’re pleased to inform you that your manuscript has been judged scientifically suitable for publication and will be formally accepted for publication once it meets all outstanding technical requirements.

Kind regards,

Ying Ma, Ph.D.

Academic Editor

PLOS ONE

Additional Editor Comments (optional):

Reviewers' comments:

Reviewer's Responses to Questions

**Comments to the Author**

1. If the authors have adequately addressed your comments raised in a previous round of review and you feel that this manuscript is now acceptable for publication, you may indicate that here to bypass the “Comments to the Author” section, enter your conflict of interest statement in the “Confidential to Editor” section, and submit your "Accept" recommendation.

Reviewer #1: All comments have been addressed

Reviewer #2: All comments have been addressed

2. Is the manuscript technically sound, and do the data support the conclusions?

Reviewer #1: Yes

Reviewer #2: Yes

3. Has the statistical analysis been performed appropriately and rigorously? 

Reviewer #1: Yes

Reviewer #2: (No Response)

4. Have the authors made all data underlying the findings in their manuscript fully available?

Reviewer #1: Yes

Reviewer #2: Yes

5. Is the manuscript presented in an intelligible fashion and written in standard English?

Reviewer #1: Yes

Reviewer #2: Yes

6. Review Comments to the Author

Reviewer #1: Dear authors,

Thanks a lot for clarification of your interesting Ms. Based on the revised version (id PONE-D-19-32967), I have decided that the manuscript can be accepted for publication in PLOS ONE. It is a very interesting manuscript.

Reviewer #2: I have no more comments as authors have addressed my all comments with proper explanations.

I suggest accept.

7. PLOS authors have the option to publish the peer review history of their article (what does this mean?). If published, this will include your full peer review and any attached files.

Reviewer #1: Yes: Tancredo Souza

Reviewer #2: Yes: Pankaj Kumar

---

## [Editor Report · Acceptance letter]

28 Jul 2020

PONE-D-19-32967R1 

Influence of Funneliformis mosseae enhanced with titanium dioxide nanoparticles (TiO2NPs) on Phaseolus vulgaris L. under salinity stress 

Dear Dr. El-Gazzar:

I'm pleased to inform you that your manuscript has been deemed suitable for publication in PLOS ONE. Congratulations! Your manuscript is now with our production department. 

Kind regards, 

on behalf of

Dr. Ying Ma 

Academic Editor

PLOS ONE